# How Teachers Deal with Cases of Bullying at School: What Victims Say

**DOI:** 10.3390/ijerph17072338

**Published:** 2020-03-30

**Authors:** Ken Rigby

**Affiliations:** Department of Education, University of South Australia, Adelaide 5001, Australia; ken.rigby@unisa.edu.au

**Keywords:** bullying, schools, teachers, surveys, interventions

## Abstract

Student victims of peer bullying (*n* = 223) in 25 coeducational Australian schools answered a questionnaire to provide accounts of how their school responded to their requests for help. In addition, respondents indicated how severely they were emotionally impacted by the bullying and whether the bullying was perpetrated by an individual or by a group. The reported outcomes from the intervention indicated that in 67% of cases the bullying stopped or was reduced. In cases where the emotional impact was reported as relatively severe, the school interventions were less successful. In addition, reportedly being bullied relatively often by groups, as distinct from individuals, was independently predictive of a less positive outcome. Among girls, but not boys, younger students reported more satisfactory outcomes. Implications are suggested for more effective interventions in cases of bullying.

## 1. Introduction: 

Schools in many countries are legally bound to take action in cases of identified bullying [1] What specific actions schools actually take in addressing cases has been described in a number of countries, drawing upon reports from school personnel in England [2] and Australia [3] and by educational psychologists in the USA [4]. However, findings based upon reports from victims of school bullying who have sought help from teachers or school counselors are notably absent. This source of information may at least be valid and can provide a complementary account of what schools do (or fail to do) to deal with cases of bullying. It can also help in identifying characteristics and experiences of victims that are associated with the reported effectiveness of teacher interventions. This article is based primarily on the reported experiences of victimized students attending schools in Australia who sought help from teachers.

Bullying has been defined broadly as the systematic abuse of power [5] Consistent with a formulation originally proposed by Olweus [6], bullying is evident when a more powerful person or group repeatedly over time acts negatively towards a targeted person who is unable to defend himself or herself adequately. Various forms of bullying have been identified, some overt, as in physical and verbal abuse; some covert, as in exclusion, rumor spreading, and the negative use of cyber technology. Victims of bullying frequently report that they are targeted in a number of these ways. According to a report by the United Nations Educational, Scientific and Cultural Organization [7]), bullying constitutes a problem present in each of the 144 countries in which student surveys have been conducted. It was estimated that worldwide approximately one child in three experiences bullying from other children at school.

This article focuses on the reported efforts of schools to address cases of bullying that come to their attention from victimized students who want their help. It does not examine the equally important proactive work that is done by schools in seeking to prevent bullying between students, for example, through education designed to develop positive social and emotional attitudes and skills in students. Such interventions may result in fewer cases taking place and improve the likelihood that case interventions may be successful [8,9]. However, schools still need to carry out reactive interventions when cases of bullying are reported to them.

The extent to which schools are effective in handling cases of bullying between students has been examined in two ways. One is based on information obtained from schools as to the success of their case interventions. In England, Thompson and Smith [2]) examined reports based on school records of outcomes; these were supplemented by information from some student interviews. It was estimated that bullying stopped in the English schools following intervention in 67% of cases; in 20% it had reduced; and in 13% it was found ineffective. From a smaller survey conducted in 25 schools in Australia it was estimated by teachers that reductions in bullying took place after the interventions in 77% of cases, with 19% remaining the same, and 4% getting worse [10]. Finally, a study conducted in Finland concluded that bullying was reduced, if not stopped, in as many as 98% of cases [11] Averaging these results provides a rough estimate of overall success in reducing, though not necessarily stopping bullying, through case interventions that approach 90%.

An alternative approach derived estimates of success from surveys of students who sought help from teachers in a number of countries, including England [12], the Netherlands [13], the USA [14] and Australia [3,15,16] Some variations in the estimates occurred, possibly due to differences in the age groups sampled, national culture, minor variations in the wording of items, and the year of testing. There is evidence that the prevalence of bullying has reduced somewhat over time in some countries [17]); therefore, precise comparisons cannot be made. However, taking victims’ claims that the bullying stopped or reduced following the intervention as a criterion of success, a rough estimate is that on average approximately 50% of interventions achieved some success. A recent study from Germany, which included student reports from victims, bullies, and other students who had witnessed bullying at their school, provided an estimate of 66% [18]. Details of these studies are given in Table 1.

### 1.1. The Nature of Interventions in Cases of School Bullying

How teachers and school counselors act when they intervene in cases of bullying may involve a variety of methods [19,20]. Their use has been described in reports provided by educational psychologists and teachers. In the USA, educational psychologists provided estimates of the actions taken by teachers to address cases of bullying. The actions perceived to be most common (agreed upon by over 90% of the educational psychologists) were: having a talk with the bullies; taking disciplinary action; increasing supervision of areas; talking with the victim; counseling the bullies; and counseling the victim [4]. In England, a survey of 1378 schools was conducted to ascertain what actions they undertook in cases of bullying [2]. In the English study specific intervention methods were stipulated. Over 90% of the schools reported that they sometimes made use of direct sanctions, such as reprimands, detentions, and suspension, but much more commonly their main strategy was the use of restorative practices, with non-confronting methods such the Support Group Method [21] and the Method of Shared Concern or Pikas Method [22] which was used in only a small minority of cases.

Studies of how students see teachers as tackling cases of bullying are rare. An exception is the survey reported in Germany [1]. In that study reports were obtained from students in 24 schools who indicated that they had witnessed a bullying incident. Of these, 71.5% were bystanders, 13.3% were described as ‘bullies’, and 15.2% were considered victims. On the basis of responses to nine questions, three forms of intervention were inferred: authoritarian-punitive (29%); supportive-individual (53%); and supportive-cooperative (18%). Arguably, students occupying the different roles would have different opportunities for providing informed judgements on what actions were taken by teachers. Being directly involved, the victims of bullying would arguably, in most instances, be in the best position to describe how their case was handled as well as whether the bullying ceased.

### 1.2. Victim-Reported Experiences and Effective Teacher Action

The success of interventions may be influenced by a number of factors. These include the intervention method adopted and applied by a teacher or counselor and its appropriateness for a given case [2]. In addition, it is proposed that some cases of bullying are generally more difficult than others to tackle effectively. This can occur when the bullying is particularly severe, such as to render the victim virtually unable to cope, even with advice and assistance from teachers and others. Where groups of students are engaged in bullying someone, the victim is more likely to feel vulnerable, as group members are able to plan, coordinate, and carry out their attacks collectively or singly in different locations. Further, interventions by the school may require working closely with a number of perpetrators, rather than just one, and with groups of students whose behavior needs to be changed. To examine the contribution of these factors—bullying severity and being bullied by a group—it is considered necessary to find out from victims of bullying how they have been affected by their experiences and the extent to which the bullying was attributed to a group as distinct from single individuals.

### 1.3. Severity

The severity of bullying is sometimes inferred from the nature of the actions taken by the perpetrator; for example, whether it involves physical assault or verbal harassment, with the former commonly regarded by teachers as more severe [23]. However, in practice, students who are bullied at school commonly claim that a variety of methods of bullying are employed, as confirmed by reports showing high levels of internal consistency when multi-item scales containing items tapping different forms of bullying are employed [24]. Further, making inferences about severity from reported behaviors ignores the fact that students differ widely in degree of resilience to stress, with some students being much less negatively affected by being bullied. Hence, in assessing severity a subjective method is preferred, as in inferring severity from reports provided by victimized students on the negative emotional impact bullying has had on them [25].

### 1.4. Bullying by Groups

It is difficult to determine by direct observation whether bullying is being perpetrated by an individual and/or by a group. It requires virtually inaccessible knowledge of whether the recorded bullying actions were due to group influence or due to individual volition. One can however obtain a measure of whether the victimized student perceives the bullying as due to a decision made entirely by an individual or as a consequence of group pressure; that is by attribution to a particular source. This approach has been employed in studies based on interviews or questionnaire data involving schoolchildren in the USA [26], Sweden [27], and Greece [28]. These studies support the view that students are more likely to report that they are bullied more often by an individual than by a group. A further study in Australia, drawing on retrospective accounts of adults regarding their childhood experiences, supports the opposite view: that victims of bullying are more likely to report being bullied by groups [29]. As the respondents in this study were students reporting on their current or recent experiences, it was expected that students would tend to report being bullied more by individuals.

### 1.5. Gender and Age

In addition to bullying severity and perceived group involvement as indicated by the victims, it was of interest to examine (and control for) the possible contribution of gender and age to intervention outcomes. It has been reported that females are more likely that males to express negative emotions when emotionally aroused, but not to experience stronger emotions as indicated by measures of physiological reaction [30]. This is consistent with the stereotype that boys may be more prone to deny or minimize negative feelings, thereby promoting a macho image of themselves [31]. However, it is possible that teachers to whom bullying is reported may take such gender differences into account when dealing with cases. Age has been suggested as a possible factor in influencing how teachers respond to cases of bullying. It was reported that bullying interventions are generally more effective with younger students [32,33]. Arguably cases of bullying among younger children can be dealt with more easily, as the bullying they experience tends to be more direct and overt [34] and consequently easier to observe and monitor.

### 1.6. Aim and Hypotheses

The aim was to describe what victimized students believed the school did after they sought help from a teacher.It was hypothesized that:
the success of the intervention, as reported by students, was inversely related to the severity of the reported negative emotional impact of the bullying;the greater the reported frequency with which the bullying was seen to be perpetrated by a group of students, the less successful the intervention would be;interventions with younger students would be associated with more positive outcomes.

## 2. Method

The method employed was a survey conducted online with a sample of Australian students attending co-educational state schools in 2015–2016. Data for the research reported in this article formed part of a larger government-funded study of the prevalence and effectiveness of anti-bullying strategies employed in Australian schools (Rigby and Johnson, 2016).

### 2.1. Ethics

Approval to conduct the research was obtained from the Ethics Committee of the University of South Australia and from each of the six educational jurisdictions in which the research was conducted. Protocol number 000032093.

### 2.2. Procedure

Contacting schools was facilitated by Principals Australia Institute (PAI) through their newsletter which went to all schools in Australia inviting them to participate. It was explained that the project was part of a government-supported project and participation of students would require the positive approval of parents or caregivers.

### 2.3. Sample

In total, 25 schools self-selected to participate in the study, including 16 primary schools, five secondary, and four combined schools, the latter catering for both primary and secondary aged students. The schools were located in the six states or territories from which permission to undertake the research was given by the relevant government jurisdiction. In total, questionnaires were answered, partially or completely, by 913 girls and 755 boys with ages ranging from 8 to 16 years. The sub-sample from which data was used in this study consisted of those students (131 girls and 92 boys) who indicated that they had sought help from a teacher over bullying and reported on its effectiveness.

## 3. Measures

### 3.1. Frequency of being Bullied

Students were provided with the following definition of bullying: “Bullying occurs when a more powerful person or group of persons repeatedly seek to upset hurt or intimidate somebody. It is NOT the same thing as occasional quarrelling between people who are about equally matched.”

Students were asked in two questions to say whether over the last 12 months they had (i) been bullied by an individual student and/or (ii) been bullied by a group of students. Categories provided were (i) never, (ii) sometimes, (iii) quite often, and (iv) very often. Responses were scored for reports of being bullied by (i) individuals and (ii) groups separately, with ‘never’ scored as 1, ‘sometimes’ as 2, ‘quite often’ as 3, and ‘very often’ as 4.

### 3.2. Severity of the Bullying

The severity of bullying was assessed by the following items: At the time you were bullied which of these words would best describe your feelings? How did you feel at the time you were being bullied? Four emotional states were indicated: angry, upset, sad, and frightened. The selection of these items was based upon research findings on the emotional impact of bullying reported by Ortego et al. In relation to each of these respondents could indicate ‘not,’ ‘a bit’, ‘quite’, or ‘very’. Reponses were score from 1 to 4 and summed to provide a measure of reported emotional impact. Its reliability as a Likert scale was confirmed, with an alpha coefficient of 0.80, based on *n* = 223.

### 3.3. Effectiveness of the Intervention

The effectiveness of the intervention was assessed by asking students who previously indicated that they had sought help from a teacher or counselor after being bullied: What happened to the bullying over the next few weeks? The response categories were: (i) it stopped, (ii) it reduced, (iii) it stayed the same, or (iv) it got worse. Responses were coded to provide a 4-point scale in the direction of successful outcomes, such that ‘it stopped was scored as 4.

### 3.4. Victims’ Perceptions of Actions taken by the School after Requesting Help

In addition, students were asked to provide information about what the teachers or counselors did after they were informed by the student of the bullying. A list of 16 possible actions was provided, to which respondents were asked to say whether or not this had been done or whether they were unsure. The list is provided in Table 2 together with student responses.

### 3.5. Data Analysis

Data were analyzed accordingly using a variety of tests: Chi-squared, related sample t-test, and multiple regression. The latter was used to assess the significance and direction of the possible influence of (i) the reported emotional impact of the bullying and (ii) the frequency of the kind of bullying (individual or group), age, and gender. Cronbach’s alpha coefficient was used to assess the internal consistency and related reliability of the Emotional Impact Scale. Missing data were excluded from each of the reported analyses conducted.

### 3.6. Demographics

Students were asked to indicate their gender (boy or girl) and give their age in years.

## 4. Results

### 4.1. Reported Frequencies of Bullying

Approximately half the respondents (*n* = 682) in the total sample reported that they had been bullied at least once during the previous 12 months. Of these, 41.1% of boys and 35.6% of girls sought help from teachers. Among these students, frequency scores were available for 209 students who provided responses relating to being bullied (i) by an individual as well as (ii) by a group. The results from the related samples’ t-tests indicated that students who sought help from teachers were more likely to have been bullied by an individual than by a group. For group bullying, the mean was 1.89 SD = 0.94; for individual bullying, the mean was 2.24, SD = 0.89, df = 208; t = 5.025, *p* = 0.000.

### 4.2. Reported Teacher Actions

The actions reported by students covered a wide range with more than 50% of students reporting that sanctions of some kind (mainly verbal reprimands) were used. In addition, teachers provided advice on how to cope with the bullying; some form of conflict resolution, like restorative practices, occurred; and following the intervention teachers monitored what happened next. Some 40% of students reported that parents of those involved in the bullying were involved.

Further details of the actions reportedly taken by the school are given in Table 2. Between 14% and 30% of students were evidently unaware of whether the school had taken certain actions.

### 4.3. Reported Outcomes for Bullying following Interventions in Cases according to Gender and Age Group

Overall, results available from students who provided answers to questions on gender, age, and reported outcomes (*n* = 223) indicated that bullying stopped or reduced in 67% of cases after seeking help from a teacher or school counselor. Outcomes for boys and girls according to age group (under 13 years and 13 years and over) are given in Table 3.

Comparisons between outcomes for boys and girls indicate non-significant differences in the distributions: Chi-squared = 1.03, df =3, *p* = 0.79. However, among girls, the distributions according to age groups were significantly different (Chi-squared = 8.16, df = 3, *p* = 0.04), with interventions being more positive for younger girls. Among boys the association with age group was not significant: Chi-squared = 5.11, df =3, *p* = 0.16.

The hypothesis that interventions with younger students would be associated with more positive outcomes following a teacher intervention was thus only partially supported, that is with respect to girls only.

### 4.4. Reported Emotional Impact and Reported Frequency of being Bullied by (i) an Individual and (ii) a Group in Relation to Intervention Outcomes

Relationships were examined between bullying outcomes and (i) reported emotional impact, (ii) reported frequency of being bullied by individuals, (iii) reported frequency of being bullied by groups, (iv) gender, and (v) age among victims of school bullying who sought help from teachers. To assess the contribution of each factor to bullying outcomes multiple regression analysis was used. The independent variables were entered together. The dependent variable—bullying outcome—was assessed on a four-point scale ranging from (1) bullying became worse to (4) the bullying stopped. To test for the variance inflation of beta coefficients a variance inflation factor (VIF) was computed for each independent variable (see Table 4).

Significant results were found, as hypothesized, for two of the independent variables: emotional impact (*p* < 0.05) and reported frequency of group bullying (*p* < 0.001) In each case the effect was negative; that is, the greater emotional impact reported by the victim, the less successful the intervention and the greater the reported group bullying, the less successful the intervention. The frequency of individual bullying, age, and gender was unrelated to student-reported outcomes.

## 5. Discussion

This is the first account of how students bullied at school reported on what teachers or counselors did to address their case. These findings complement those provided in earlier studies in which information was derived from school authorities.

As reported by schools in studies reviewed in Section 1, a wide variety of methods have been employed in dealing with cases of bullying. These include direct confrontation and punishment of perpetrators, providing advice to the victim on how to cope, meetings with bullies and victims, such as in restorative practices, involvement of the parents of the students involved, and classroom meetings. As in some previous studies, the results from the student survey suggest that sanctions or disciplinary measures directed towards the perpetrators were the most common intervention methods [36,37], rather than the supportive-individual strategies (gathering information, talking to involved pupils, supporting them emotionally) reported as most commonly used by teachers in the German study [18]. It was however evident from the answers provided by students to questions of what actions were taken by the school that in many cases the students did not know, possibly because the action (e.g., meeting with the parents of the perpetrators) was not disclosed to them.

In general, the outcomes of the interventions as reported by students were less positive than those reported by schools, as reviewed earlier. In the most recent study of the outcomes from teacher interventions in Australian schools, as provided by teachers (3) improvements did not occur in approximately 23% of cases Results from the student survey indicate that no improvement was evident in 33% of cases. This study suggests that the failure to address such cases successfully may be related to two hypothesized factors: (i) the severity of the bullying experienced, as inferred from reports of the emotional impact, and (ii) the experience of being bullied relatively frequently by a group(s) of students. The hypothesis that outcomes from interventions would be more positive for younger students, supported in earlier studies, was supported only for girls.

The findings that outcomes were less positive in cases of relatively high negative emotional impact and in cases of frequent group bullying may help to explain why some interventions were unsuccessful. Children who become highly anxious or depressed—and in some extreme cases traumatized—by severe bullying (see [28]) are less able to resist attacks from peers by applying appropriate social and interpersonal skills. They may also find themselves despised, isolated, and without helpful peer support [38,39] which may further contribute to their vulnerability. Assisting such students through what has been termed ‘strengthening the victim’ [19] is difficult to achieve without considerable perseverance and skill. It is evident from the student survey that many teachers provide advice on how a victimized student can cope more effectively, but in the case of severely emotionally disturbed students the difficulty may be too great, and the use of punishment of the bully or bullies may sometimes make matters worse, especially when monitoring of further bullying is inadequate.

Although group bullying—or the perception of being bullied by a group—was found to be significantly less common than being bullied by an individual, the less successful outcomes obtained for group bullying suggest that dealing with such cases presents greater difficulties for schools. Generally, it involves working with a number of perpetrators who tend to influence each other. Directly confronting a group of students thought to be the perpetrators or their supporters often at different times may entail mistakenly accusing students who have played little or no part in the bullying; this may stir up resentment and even strengthen the feeling of pride and mutual loyalty shared by the members of a group who resolve to continue bullying in less overt ways. Alternative and special ways of dealing with cases of group bullying, such as the Support Group Method [21]) and the Method of Shared Concern are not well known and are practiced little in schools (2,3).

### 5.1. Implications for Interventions in Cases of Bullying

Results from this study suggest that the failure of some schools to address cases of bullying may be related to the severity of the bullying and the frequency with which the bullying is seen as being undertaken by groups. Given these findings, how can schools improve their rates of success?

The first step lies in assessing the severity of the bullying and recognizing that high levels of severity, as indicated by the negative emotional impact on the child, may call for sustained empathic concern and appreciation of the difficulties that confront the child in coping with the bullying. This may involve referral in the most extreme cases to professional services outside the school, which according to the victimized students occurs in 12% of cases, and/or coaching the student to develop interpersonal skills to effectively deal with being bullied, such as ‘fogging’ (19). Working closely with parents and offering them emotional support and advice can also make an important contribution [40].

To deal more effectively with group bullying schools may opt to adopt methods of intervention designed for such cases. This may entail applying the Support Group Method, according to which victims are first interviewed to gather detailed knowledge of how they have been affected by the bullying and who are the known perpetrators. Subsequently this knowledge is shared with the ‘bullies’ at a meeting that includes students who are expected to provide support for the victim and influence the bullies to do likewise. High levels of success have been reported by independent assessors of the method’s effectiveness [41]. Another intervention method designed for use with group bullying, known as the Method of Shared Concern, or Pikas Method after its originator, has been reported as highly effective [42,43]. With this method a comprehensive approach is adopted beginning with one-on-one interviews with the suspected bullies (no one is accused) followed by a meeting with the targeted student; then a meeting with all the suspected bullies takes place to devise a positive proposal to be presented as a basis for a negotiated solution, and if possible, agreed upon in a final group meeting involving the victim. The limited amount of education and training for teachers and school counselors is forestalling the use of this and other methods in many schools [10,23].

### 5.2. Strengths and Limitations

The primary strength of this study is that it provides, for the first time, victimized students’ accounts on what schools do to help them when they seek help from teachers in response to bullying by other students and the success of the school intervention. As such, it complements other studies based on reports provided by schools. This study identifies two significant factors—the severity of the bullying and the frequency of being bullied by a group—as associated with relatively unsuccessful outcomes from interventions. It is acknowledged that many other factors may affect the effectiveness of teacher interventions. These include personality attributes, such as personal aggressiveness, in part genetically determined [44], and home backgrounds that may result in some children having inadequate social skills [40]. In this study bullying was assessed in a general sense, without considering specific forms of bullying such as cyberbullying and how well each form is addressed by schools. Further, it does not purport to provide an objective assessment of how particular students were bullied. It relies upon attributions of the source of the bullying. It does not examine whether the person seeking help has in fact engaged in bullying others, as some have claimed [45], a factor that may affect a teacher’s motivation to intervene. This study was limited in that it examined the possible effects of chronological age, between 8 and 16 years only, and did not take into account physical, social, and cognitive developmental stages that could influence teacher interventions. A final limitation lies in not relating outcomes to particular schools or types of schools, thereby not taking into account variations between schools in outcomes and related factors, such as the anti-bullying policy adopted by the school. Further research is needed to determine whether the findings from this study are limited to the Australian context.

## 6. Conclusions

According to students attending schools in England, USA. Netherlands, Germany and Australia, teacher interventions to stop bullying from continuing are often unsuccessful. Research undertaken in Australia has identified two factors that are related to negative outcomes, namely the severity of the reported emotional distress experienced by the bullied child and the bullying being perpetrated relatively frequently by groups rather than by individuals. Appropriate means of intervening in such cases need to be more widely recognized and applied.

## Figures and Tables

**Table 1 ijerph-17-02338-t001:** Outcomes from student surveys: percentages reported.

Student Reports	Reported Effects on Bullying				
	Stopped	Reduced		No Change	Got Worse
Shu and Smith, 2000	26		29	28	16
Rigby, 1998		**49**		43	8
Rigby and Barnes, 2002		**42**		39	18
Fekkes, Pijpers and Verloove-Vanhorick, 2005		**49**		34	17
Davis and Nixon, 2011		**34**		37	29
Rigby and Johnson, 2016	29		40	23	8
Wachs et al., 2019 *	22		44	30	4

Note: Bold data: combine results for ‘stopped’ and ‘reduced’ as only these figures were available. * Results for Wachs et al. excluded the students (17%) who reported that they did not know the outcome of the intervention.

**Table 2 ijerph-17-02338-t002:** Frequency of reported actions taken by schools in relation to reported bullying.

Reported Teacher Action	Yes	DK
**Confront and apply sanctions to the bully**		
The teacher told the bully or bullies to stop bullying me	63.5	17.8
A teacher got the bully or bullies to apologize	60.0	14.9
The bully was given a warning	55.7	26.6
The teacher deprived the bully of privileges at school	25.4	27.5
The bully/bullies were given a detention	22 7	23.4
The bully was made to do community work	22.4	21.4
The bully was suspended from school	17.3	16.8
The bully was excluded from school	9.6	13.7
**Helping the victim to cope**		
A teacher advised me on what I could do to stop the bullying	62.7	15.7
**Engaged with parents**		
The school got in touch with parents of the student(s) bullying me	41.7	26.0
A teacher talked to my parents about what was happening	40.7	20.1
The school suggested my parents get in touch with the bully’s parents	19.9	29.9
**Undertook conflict resolution**		
A teacher met with me and the bully to sort things out together	51.5	17.2
The school arranged meeting with a student mediator	14.5	29.5
**Sought help from outside the school**		
Arranged for help from outside school, e.g., a psychologist	12.2	17.3
The police were informed	6.2	21.6
**Monitored student behavior**		
The teacher kept an eye on things for the next few weeks	52.3	25.1
**Working with students**		
The teacher spoke with the class to get their help	42.0	17.6

**Note:** Items have been slightly abbreviated from those given in full in Rigby and Johnson (2016). ‘DK’ indicates percentages of students indicating they did not know whether this action had taken place. Not all students who sought help from a teacher answered every question. Respondents to the questions ranged from 192 to 204.

**Table 3 ijerph-17-02338-t003:** Reported outcomes of school interventions in relation to gender and age group.

		Bullying Stopped	Bullying Reduced	Bullying Stayed the Same	Bullying Got Worse	Total *n*
Boys						
	Young	19 (29.7)	21 (32.8)	18 (28.1)	6 (9.4)	64
	Older	5 (17.9)	16 (57.1)	6 (21.4)	1 (3.6)	28
Girls						
	Young	33 (38.4)	32 (37.2)	14 (16.3)	7 (8.1)	86
	Older	8 (17.8)	19 (42.2)	15 (33.3)	3 (6.7)	45

Note: Age groups defined as young if under 13 years; older if 13+ years.

**Table 4 ijerph-17-02338-t004:** Results for regression analysis: predicting success of teacher interventions in cases of bullying.

Outcome of Teacher Interventions				
	Beta	*t*	*p*	VIF
**Independent Variables**				
Reported emotional impact	−0.174	−2.699	0.014	1.234
Frequency of individual bullying	−0.115	−1.584	0.115	1.317
Frequency of group bullying	−0.218	−4.147	0.000	1.257
Age (in years)	−0.100	−1.547	0.123	1.033
Gender (Boy = 1; Girl = 2)	+0.070	+1.072	0.285	1.061

Total R = 0.458, R^2^ = 0.190; ANOVA: df = 5197; F = 10.451, *p* < 0.001. Note: Variance inflation factor (VIF) values associated with the independent variables ranged from 1.033 to 1.317. These results imply that collinearity only had minimal effects on the magnitude of the reported beta coefficient [35].

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
