# Peer review of "How Teachers Deal with Cases of Bullying at School: What Victims Say"

_ijerph, 2020, doi:10.3390/ijerph17072338_

Round 1

Reviewer 1 Report

The manuscript titled "How teachers deal with cases of bullying at school:
what victims say" investigates how schools / teachers have helped or responded to students help requests regarding bullying.

Introduction:

Authors briefly introduce the theme and its necessity to be investigated, informations regarding students who have asked for teachers / school hel are usually not present in bullying studies.

Also the importance of understand and study bullying is clear defined. Statistics comparing incidence of improvement (or not) are presented.

Table 1: please leave a space between the last paragraph and table name.

In table 1 is difficult to understand which numbers refers to the columns, please edit the table to be clearer. Numbers can not be readen correctly in the box as well.

The introduction section is very rich on information but a bit to long, authors could try to summarize all informaiton presented.

Line 149 "bullying interventions are generally more effective with younger students" what does this means? Are younger students 8 years old, or 10 or 12? Please be more specific. Since authors are arguing on the possible age influence this information is relevant.

Please write the hypotheses (from line 156) with the same font / font size.

Method:

Participantes selection is ok. How did authors handle missing information (since it was reported the questionnaires were only partially answered line 184)?

Authors have designed their own questionnaire, no validated questionnaire was used - why? Is there any of it?

The statistical / mathematical software used for the analysis was not reported.

Results:

The sample sizes are confusing, in abstract and section 3.2 authors reported a sample size of N=223 but on section 4.1 they reported a sample size of N=350 - where does numbers come from? Please specify this.

Authors present the Wilcoxon test regarding a comparison of being bullied by an individual or b a group having resulted a significant result of being bullied by an individual but he Figure 1 is a comparison between gender - this is confusing.

Please edit all tables to have also the same font / font size as text, as also allign its colums. In table 2 what DK (on the side of YES) means?

Regarding p-values: please specify the excat values not only p> or p< 0.5.

Authors reported using multpile regression but since the dependent variable is orderd the correct method is the multinomial ordered regression. Please readjust the analysis.

Discussion:

Authors discuss once more their findings and relates to some bibliography. A very clear and didactic discussion is presented.

Author Response

Introduction:

Authors briefly introduce the theme and its necessity to be investigated, informations regarding students who have asked for teachers / school hel are usually not present in bullying studies.

Also the importance of understand and study bullying is clear defined. Statistics comparing incidence of improvement (or not) are presented.

Table 1: please leave a space between the last paragraph and table name.

Done

In table 1 is difficult to understand which numbers refers to the columns, please edit the table to be clearer. Numbers can not be readen correctly in the box as well.

Table has been modified as far as possible, as required.

The introduction section is very rich on information but a bit to long, authors could try to summarize all informaiton presented

Comment noted and some shortening attempted However the detail is highly pertinent to the article..

.

Line 149 "bullying interventions are generally more effective with younger students" what does this means? Are younger students 8 years old, or 10 or 12? Please be more specific. Since authors are arguing on the possible age influence this information is relevant.

What is meant by younger is clarified – i.e of a lower age in years. There are no research grounds of which I am aware for supposing that belonging to an particular age category has implications for the effectiveness of interventions, although being relative young has been suggested as a factor and is investigated. Agreed that developmental  stages - physical, social, emotional, cognitive -  could be relevant and categorization is possible  This is  now acknowledged in the limitations section  

Please write the hypotheses (from line 156) with the same font / font size.

Done

Method:

Participantes selection is ok. How did authors handle missing information (since it was reported the questionnaires were only partially answered line 184)?

This is now made clear

Authors have designed their own questionnaire, no validated questionnaire was used - why? Is there any of it?

No pre-existing measure of frequency of (i) individual and (ii) group bullying could be found. The choice of items for emotional impact scale was based on research reported by Ortega et al., as mentioned. The reliability of this measure was tested and reported in the article submitted.  

The statistical / mathematical software used for the analysis was not reporte.

SPSS was used as now reported

Results:

The sample sizes are confusing, in abstract and section 3.2 authors reported a sample size of N=223 but on section 4.1 they reported a sample size of N=350 - where does numbers come from? Please specify this

Correct sample sizes are now in place. Error corrected.

.

Authors present the Wilcoxon test regarding a comparison of being bullied by an individual or b a group having resulted a significant result of being bullied by an individual but he Figure 1 is a comparison between gender - this is confusing.

The test is applied to results from students who were bullied and reported on both individual and group frequency. In the revision the related samples t =test has now been used.. The Figure- which is not relevant to the analysis- has been deleted Its removal does affect the rest of the article.

Please edit all tables to have also the same font / font size as text, as also allign its colums. In table 2 what DK (on the side of YES) means?

Table 2 adjustments made. ‘DK’ is explained as ‘ Don’t Know.’

Regarding p-values: please specify the excat values not only p> or p< 0.5.

Exact p values have been inserted throughout.

Authors reported using multpile regression but since the dependent variable is orderd the correct method is the multinomial ordered regression. Please readjust the analysis.

The thoughtful suggestion that multinomial regression should be employed in place of multiple linear regression has been considered. Unfortunately the hypotheses related to predictions of positive or negative outcomes of interventions. Multinomial analysis would indicate whether given factors discriminate significantly between the four different outcomes as categorized, but not answer the question of whether each independent factor is related to a positive or negative intervention effect, as required by the hypotheses

It is appreciated that the linear regression analysis employed assumes a continuity in the scaling of the dependent variable. In this inquiry, as you see, the reported effect of intervention is coded 1 to 4 in a positive direction. Whether the scaling should be regarded as ordinal or interval and therefore requiring different forms of analysis is controversial. As discussed by Norman (2010), in practice in the social sciences the use of such a measure as a dependent variable in parametric analyses such as regression analysis is generally allowed, especially if the distribution of scores is fairly close to normal. This is the case with the dependent variable used in this analysis. On balance, the analysis employed in the paper appears justified in this instance.

Norman, G. (2010) Likert scales, levels of measurement and the “laws” of statistics. Advances in Health Sciences Education 15 (5):625-32.

..

Discussion:

Authors discuss once more their findings and relates to some bibliography. A very clear and didactic discussion is presented.

Thank you

Reviewer 2 Report

This is an interesting paper focusing on an important topic. I think some relevant issues should be addressed before the manuscript can be considered for publication. In particular:
- throughout the paper, I see some problems in the formatting of the text and tables, which sometimes make the article difficult to read.
- the introduction section, although potentially useful as a review of relevant literature in the field, is too long and should be reduced.
- Although very long, the introduction section does not encompass any description of the cyberbullying phenomenon that, instead, is unfortunately very common among youths. I think that this point should be addressed.
- I suggest that the authors clarify why the study is being proposed for publication after five years from the sample gathering.
- I see no reference to the fact that the age range (8-16 years) is very wide and that it encompasses at least 3 developmental stages: childhood, early adolescence, and adolescence. I think that the manuscript should be more developmentally oriented and should specify why bullying can be particularly frequent among adolescents. Empathy as a protective factor and altruistic help should be considered (why bystanders often do not intervene?) and relevant references should be added (Paciello, Fida, Cerniglia, Tramontano, Cole, 2013)

- Callous-unemotional traits could be associated with bullying (Milone, Cerniglia, Cristofani, (...), Simone, Muratori, 2019)

- The authors should clarify why this study did not tap any possible psychological and family issues associated with bullying and aggressive behaviors (psychopathological risk in the bully and in the victim, family factors, etc.). This should be acknowledged at least in the limitations. The reader could have the impression that this phenomenon takes place without any precursor or predictive factor. Instead, a very large literature have focused on factors fostering bullying.
- The authors should clarify why they chose mostly ad hoc measures instead of choosing validated tools to tap the study variables.
- another limit is that the authors do not mention the fact that, as it is widely known, bullies are often also victims (and vice versa). Relevant references should be added.
- The sampling procedure is not entirely clear. At line 184 the authors mention 913 girls and 755 boys, however afterward the final sample appears much smaller. Why?

Author Response

This is an interesting paper focusing on an important topic. I think some relevant issues should be addressed before the manuscript can be considered for publication. In particular: throughout the paper, I see some problems in the formatting of the text and tables, which sometimes make the article difficult to read.

Formatting has been adjusted - though it appears that what is Ok at my end seems to get distorted in transmissiion. I hope this can be fixed,

- the introduction section, although potentially useful as a review of relevant literature in the field, is too long and should be reduced

Appreciated, but hard to reduce as this background is all needed and as far as intervention outcomes are concerned has not been brought together before..

-Although very long, the introduction section does not encompass any description of the cyberbullying phenomenon that, instead, is unfortunately very common among youths. I think that this point should be addressed.

I have to disagree here. The assessment of bullying was a general one and in fact nearly all students who are cyberbullied are also being bullied in other ways. Reporting being cyberbullied is part of the syndrome, no more nor less than other forms such as exclusion and malicious rumour spreading. The hypotheses refer to bullying in general.

I suggest that the authors clarify why the study is being proposed for publication after five years from the sample gathering.

Four years. Data for this study derive from a questionnaire used as part of a much larger study which dealt with a wide range of issues. Analyses and writing papers has been delayed as different issues were being addressed.

- I see no reference to the fact that the age range (8-16 years) is very wide and that it encompasses at least 3 developmental stages: childhood, early adolescence, and adolescence. I think that the manuscript should be more developmentally oriented and should specify why bullying can be particularly frequent among adolescents.

Empathy as a protective factor and altruistic help should be considered (why bystanders often do not intervene?) and relevant references should be added (Paciello, Fida, Cerniglia, Tramontano, Cole, 2013)- Callous-unemotional traits could be associated with bullying (Milone, Cerniglia, Cristofani, (...), Simone, Muratori, 2019)

- The authors should clarify why this study did not tap any possible psychological and family issues associated with bullying and aggressive behaviors (psychopathological risk in the bully and in the victim, family factors, etc.). This should be acknowledged at least in the limitations. The reader could have the impression that this phenomenon takes place without any precursor or predictive factor. Instead, a very large literature have focused on factors fostering bullying.

These suggestions are welcome, but their inclusion would have resulted in extending the article enormously. A paper on the challenges of intervening in cases of children at different developmental stages, as distinct from chronological stages would be worth attempting, together with a range of personality types and social backgrounds but is  beyond the scope of this paper. Some acknowledgement of other factors is now included as limitations. Incidentally, the view that bullying can be (is?) particularly frequent among adolescents is not supported in numerous research papers. It is more frequent among younger students, and usually of a somewhat different nature

The authors should clarify why they chose mostly ad hoc measures instead of choosing validated tools to tap the study variables.

There does not appear to be any measure available for assessing the frequency of (i) being bullied by an individual and (ii) being bullied by a group. Single item measures were devised. The measure of emotional impact was modeled on the work of Ortega et al., which resulted in the selection of items. The reliability of the 4-item measure is given as .80, which supports its use.

.
- another limit is that the authors do not mention the fact that, as it is widely known, bullies are often also victims (and vice versa)

This may help to explain why teachers are sometimes reluctant to help children who claim to be victims and have provoked the ‘bully.’ Worth mentioning as it is directly relevant to why interventions are quite often not successful. Therefore added as a limitation. .

Relevant references should be added.
- The sampling procedure is not entirely clear. At line 184 the authors mention 913 girls and 755 boys, however afterward the final sample appears much smaller. Why?

The sub-sample from which data was used in this study consisted of those students who indicated that they had sought help from a teacher over the bullying and reported on its effectiveness, that is 131 girls and 92 boys This is I think now made clear.

Submission Date

01 March 2020

Round 2

Reviewer 1 Report

Authors have either explained or changed sections in text with were confusing. Tables were adjusted and statistical informations better reported.

Reviewer 2 Report

Several of the suggestions have not been addressed, however, the reasons for this have been motivated by the authors and I think their explanations are acceptable.